# In Vitro and In Silico Evaluation of Red Algae *Laurencia obtusa* Anticancer Activity

**DOI:** 10.3390/md21060318

**Published:** 2023-05-24

**Authors:** Jéssica Raquel Borges Monteiro, Ricardo Pereira Rodrigues, Ana Carolina Mazzuco, Rita de Cassia Ribeiro Gonçalves, Angelo Fraga Bernardino, Ricardo Machado Kuster, Rodrigo Rezende Kitagawa

**Affiliations:** 1Graduate Program of Pharmaceutical Sciences, Federal University of Espirito Santo, Vitoria 29047-105, Brazil; jessica_rbm@yahoo.com.br (J.R.B.M.); ricardopereira.pr@gmail.com (R.P.R.); kusterrm@gmail.com (R.M.K.); 2Department of Oceanography and Ecology, Federal University of Espirito Santo, Vitoria 29075-910, Brazil; ac.mazzuco@me.com (A.C.M.); angelofraga@gmail.com (A.F.B.); 3Graduate Program of Chemistry, Federal University of Espirito Santo, Vitoria 29075-910, Brazil

**Keywords:** *Laurencia*, seaweed, gastric cancer, molecular docking

## Abstract

Studies estimate that nearly 2 million new cases of gastric cancer will occur worldwide during the next two decades, which will increase mortality associated with cancer and the demand for new treatments. Marine algae of the *Laurencia* genus have secondary metabolites known for their cytotoxic action, such as terpenes and acetogenins. The species *Laurencia obtusa* has demonstrated cytotoxicity against many types of tumors in previous analyses. In this study, we determined the structure of terpenes, acetogenins, and one fatty acid of *Laurencia* using mass spectrometry (ESI-FT-ICR/MS). In vitro cytotoxicity assays were performed with adenocarcinoma gastric cells (AGS) to select the most cytotoxic fraction of the crude extract of *L. obtusa*. The Hex:AcOEt fraction was the most cytotoxic, with IC_50_ 9.23 µg/mL. The selectivity index of 15.56 shows that the Hex:AcOEt fraction is selective to cancer cells. Compounds obtained from *L. obtusa* were tested by the analysis of crystallographic complexes. Molecular docking calculations on the active site of the HIF-2α protein showed the highest affinity for sesquiterpene chermesiterpenoid B, identified from HEX:AcOEt fraction, reaching a score of 65.9. The results indicate that *L. obtusa* presents potential compounds to be used in the treatment of neoplasms, such as gastric adenocarcinoma.

## 1. Introduction

Cancer is the main problem in public health worldwide. The International Agency for Research on Cancer (IARC) estimated about 1 million new cases of stomach cancer in 2020, possibly reaching almost 2 million in 2040. Mortality also increases. The number of fatal victims of stomach cancer was about 700 thousand million in 2020, and 1.3 million people are expected to die of this type of cancer worldwide in 2040 [1].

Compounds extracted from marine organisms play an important role in many specific cell processes, including apoptosis, angiogenesis, migration, and invasion [2,3]. Red algae are one of the main producers of secondary metabolites in the marine environment, compounds that are often identified with a potential for industrial and medical applications [4,5]. Among them, macroalgae of the genus *Laurencia* are a model for the discovery of new bioactive compounds [6,7]. The natural products found in the *Laurencia* are mostly sesquiterpenes, diterpenes, and acetogenins, and are usually halogenated. These compound classes are known for their cytotoxic and anti-inflammatory activities [7,8]. The cytotoxic and gastroprotective activities of the *Laurencia obtusa* are well known [6,7,8,9], but have not yet been fully elucidated. Terpenes could activate the extrinsic pathway of apoptosis, decrease cyclins-dependent kinases (CDKs) and cyclins that are needed for the proper function of the cell cycle, increase pro-apoptotic factors such as Bak, Bax, and caspase-9, decrease anti-apoptotic factors such as Bcl-xl, and inhibit the HIF (hypoxia-inducible factor) family, which is responsible for the cancerous tissue oxygen demand [10,11,12,13,14].

HIF-2α is important in metastasis: in normoxia, prolyl hydroxylases’ (PHDs) oxygen dependents are responsible for the recognition of the factor by von Hippel–Lindau protein, a tumor suppressor that initiates the proteasomal degradation of the factor. However, these processes are not observed under hypoxia due to the lack of oxygen to activate the PHDs. As a result, HIF-2α accumulates and translocates to the cell nucleus, dimering with the aryl hydrocarbon receptor nuclear translocator (ARNT), also known as HIF-2β, forming an active transcription factor complex [15,16]. Therefore, the inhibition of HIF-2α could be a new strategy of treatment for many types of cancer. The screening of marine natural products is limited, since species could have different secondary metabolite profiles or not produce them in a sufficient quantity. An alternative way to overcome these limitations is virtual screening with a bioinformatics tool capable of predicting the interaction of biological targets and ligands that give crucial information about biological activities, then filtering these compounds to a reduced number to allow them to be studied [17,18].

This study aimed to screen for compounds that may be responsible for *L. obtusa* cytotoxicity against a gastric adenocarcinoma cell line and to identify potential mechanisms for their use in new cancer treatments.

## 2. Results

### 2.1. MTT-Tetrazolium Assay

The MTT-tetrazolium assay was carried out with the ethyl acetate (AcOEt), butanol (BuOH), and water (H_2_O) fractions. AcOEt and BuOH fractions have shown cytotoxicity at 100 µg/mL, such as the standard cisplatin; however, the AcOEt fraction showed greater cytotoxicity than the other fractions at the concentration of 10 µg/mL (Figure 1).

Furthermore, due to the greater cytotoxicity of the AcOEt fraction, different subfractions were prepared: hexane: ethyl acetate 3:1 (Hex:AcOEt), ethyl acetate: hexane 3:1 (AcOEt:Hex), ethyl acetate (AcOEt’), ethyl acetate: methanol 1:1 (AcOEt:MeOH), and methanol (MeOH). The Hex:AcOEt fraction was the most cytotoxic, being more cytotoxic than cisplatin at concentrations of 10 and 100 µg/mL (Figure 2).

A new cytotoxicity assay was performed with the Hex:AcOEt fraction in five different concentrations, ranging from 1 to 100 μg/mL, to estimate its IC_50_. Afterwards, the IC_50_ was determined at 9.23 μg/mL.

A cytotoxicity assay with a RAW cell line (normal cell line) was also performed with the Hex:AcOEt fraction to estimate the selectivity index (SI). The IC_50_ was 143.20 μg/mL. The SI was estimated by the ratio of RAW/AGS IC_50_. The SI value was 15.56, indicating high selectivity for tumor cells line (AGS) [19].

Based on these results, the Hex:AcOEt fraction was analyzed by electrospray ionization—mass spectrometry (ESI-MS) and subjected to docking calculations.

### 2.2. Structure Determination

The mass spectra of the Hex:AcOEt fraction in negative and positive ion modes of ESI-MS are shown in Figure 3 and Figure 4. The suggested compounds are described in Table 1 and Table 2. The most abundant peak at *m*/*z* 255.23298 corresponds to a phthalate, which does not show any correlation with the sample.

### 2.3. Molecular Docking

The compounds found in the Hex:AcOEt fraction were tested on the active site of HIF-2α using the validated score function determined by redocking. Each of them generated ten poses. The five best scores are presented in Table 3.

The interactions of chermesiterpenoid B, 11,12-dihydro-3-hydroxyretinol, and 7-hydroxy-10-dehydroxydeacetyldihydrobotrydial-1(10),5(9)-diene with the main amino acids in the active site of the protein overlaid with the ligand reference 0X3 are shown in Figure 5.

## 3. Discussion

The cytotoxicity results suggest that *L. obtusa* has a potential for cytotoxic activity against AGS cells, and this activity is similar to the cytotoxic standard cisplatin; Hex:AcOEt was the most cytotoxic fraction of the crude extract. The values of SI show that the Hex:AcOEt fraction is selective for the AGS cells, once the selective system has SI > 10 [19]. The ESI-MS results show that sesquiterpenes, diterpenes, and at least one acetogenin and one fatty acid could be present in the extract of *L. obtusa*. This is the first time these compounds have been reported in *L. obtusa*, except for brasilenone.

In the most cytotoxic fraction (Hex:AcOEt), the spectrum obtained through ESI-MS analysis suggests the presence of sesquiterpenes and diterpenes. Among them, a drimane-type sesquiterpenoid, usually obtained from the endophytic fungus *Pestalotiopsis* sp., is also very common in marine organisms [31]. This specific type of sesquiterpenoid is known by its cytotoxic activity against different tumoral cell lines. The drimane-type sesquiterpenoids, such as polygodial, warburganal, and muzingadial, have already shown cytotoxicity against MV4-11 and THP-1 (leukemia), Sk-Mel29 (melanoma), and LN-229 (gliobastoma) [32]. Merulinol F and laureacetal C are chamigrene-type sesquiterpenoids. Merulinol F is obtained from the endophytic fungus XG8D, which is widely associated with marine organisms, and the laureacetal C is isolated from the red algae *Laurencia nipponica*. This type of sesquiterpene is very common in the *Laurencia* genus and has already demonstrated cytotoxicity against the KATO-3 cell line (gastric cancer) [21,33]. Trichothecene 1 is a sesquiterpene produced by the endophytic fungus *Trichoderma brevicompactum*, which is also found in the red algae *Mastophora rosea*. Trichothecenes have demonstrated cytotoxicity against tumoral cell lines, such as HCT-116 (colon cancer), PC-3 (prostatic adenocarcinoma), and Sk-Hep-1 (liver adenocarcinoma) [22]. The eusdesma-4(15),11-dien-5,7-diol is a eudesmane-type sesquiterpene widely found on *L. obtusa* and *L. nipponica*, with cytotoxic activity against the MCF-7 cell line (breast cancer) [34]. The diterpene 11,12-dihydro-3-hydroxyretinol, found in *Laurencia okamurae* and *L. nipponica*, is a retinane-type diterpene, but, to the best our knowledge, no evidence of its cytotoxic activity has been found [23]. In addition, the sesquiterpenes 7-hydroxy-10-dehydroxydeacetyldihydrobotrydial-1(10),5(9)-diene, 5,9-dihydroxy-1(6)-brasilen-7-one, and chermesiterpenoid B also have no previous evidence of citotoxycity in the literature. Acetogenins such as laurefurenyne B are also common in the *Laurencia* genus. Laurefurenynes have already demonstrated cytotoxic activity against H-116 (colon cancer) and H-125 (lung cancer) [30].

Beyond terpenes and acetogenins, ESI-MS analysis also indicated the presence of one fatty acid: eicosapentaenoic acid (EPA). Common in marine organisms and part of the composition of food supplements, this polyunsaturated fatty acid has demonstrated the ability to prevent many types of cancer such as leukemias, breast cancer, colon cancer, prostate cancer, and melanomas [35]. During the carcinogenesis process, EPA acts like an antioxidant, suppressing the inducible nitric oxide synthase (iNOS) and, consequently, the production of reactive nitrogen species (RNS), preventing damage to DNA that could culminate in cancer development. During cancer progression, EPA can decrease expression of important proteins such as HIF-1α and VEGF, which are very important in the metastasis process and stimulate apoptosis by caspase activation [35].

Nowadays, the treatment for metastatic gastric cancer is based on palliative systemic chemotherapy that could extend the patient’s life for at least six months. The angiogenic activity inside the tumoral microenvironment is regulated by the HIF family, which indicates that the control over these factors is an important factor to decrease the effects of pathological angiogenesis [36,37]. In 2021, the Food and Drug Administration (FDA) approved a medicine capable of inhibiting HIF-2α for the treatment of renal cancer associated with von Hippel–Lindau syndrome. The MK-6482 (previously named PT2977) binds to HIF-2α and creates a steric hindrance that interrupts its dimerization with ARNT, inhibiting its transcriptional activity [15,16,38,39]. According to Kubinyi (2002), the structural similarity of molecules cannot be objectively defined: chemically similar compounds can have different biological activities, and the same ligand can interact with different receptors. The biological activity will depend on its 3D structures and its interaction with the active site of the biological target [40]. The 0X3 used in docking calculations is also a ligand capable of interrupting the transcriptional activity of HIF-2α (IC_50_ = 0.49 µM by scintillation proximity assay and IC_50_ = 0.01 µM by luminescent proximity assay) [41,42]. Because molecules with similar structures could have similar biological activity, and the score estimates the affinity of the ligand orientation in the active site, we can expect that the molecules selected by ESI-MS analysis with a high score can partially inhibit HIF-2α, similar to 0X3, the reference ligand [43].

The molecular docking performed with the Hex:AcOEt compounds suggests a binding affinity, and interaction with key residues at the binding site of HIF-2α, the biological target. All four terpenes with the best ranking score present a C_6_ ring, and the acetogenin laurefurenyne B presents a terminal triple bond similar to MK-6482. The three compounds with the highest score in docking calculations—chermesiterpenoid B, 11,12-dihydro-3-hydroxyretinol, and 7-hydroxy-10-dehydroxydeacetyldihydrobotrydial-1(10),5(9)-diene—are compounds that present no evidence of cytotoxic activity in the literature. The results of this study suggest that, according to cytotoxicity assay and molecular docking, these three compounds have the potential for cytotoxic activity against AGS cells and this activity may occur by the inhibition of HIF-2α, as proposed by the docking study.

The fragmentation of peaks found in ESI-MS would be essential to subsequently improve the understanding of the results, as well as the use of other identification techniques such as nuclear magnetic resonance (NMR), determination of the pharmacophore of the target protein, and Western blotting analysis, flow cytometry, or immunoassays to confirm the mechanism of action.

## 4. Materials and Methods

### 4.1. Seaweed Collection and Identification

The seaweed was sampled manually in shallow water (0.5 m depth) during low tide in the reefs of Gramuté beach, located within the Santa Cruz Marine Wildlife Refuge, Espirito Santo State, eastern coast of Brazil (Latitude 19°58′22″ S, Longitude 40°8′12″ W). It is a tropical region (average air temperature of 25 °C, sea surface temperature of 26 °C) with abundant rocky reefs and rich macroalgal beds that extends from the intertidal to the continental shelf boarder [44,45]. Gramuté reefs and the associated seaweeds host and serve as nursery ground for a diverse assemblage of macroinvertebrates, and have been monitored monthly by the Long-term Ecological Research Program Coastal Habitats of Espírito Santo since 2017 [44,46,47]. These seaweed beds show marked monthly variability in composition and abundance. *Laurencia obtusa* is commonly found during the summer and fall months. For these analyses, the seaweed was sampled on 7 April 2020. After sampling, one algal frond was stored in 70% ethanol solution (voucher number PELDHCES-LAU0001) in the Benthic Ecology Laboratory’s collection, at the Federal University of Espirito Santo. The species occurrence record was registered in the Ocean Biodiversity Information System data repository [48].

### 4.2. Extract Preparation

The seaweed was washed and then dried at 20 °C room temperature for seven days. After this time, 95 g of the seaweed was scraped and subjected to maceration process with dichloromethane:methanol (2:1) for 14 days. The extract was filtered and concentrated in a rotary evaporator below 60 °C until the extract was completely dry. The final yield was 9.08%.

### 4.3. Fractionations

Liquid–liquid partition was used for the fractionation of the crude extract with ethyl acetate: water (1:1) and butanol: water (1:1), obtaining the fractions ethyl acetate, butanol, and water. The organic fractions were concentrated in a rotary evaporator until completely dry. The aqueous fraction was dried using a lyophilizer. Afterward, a bioguided study was conducted for cytotoxic activity, and the most active fraction was selected for a new fractionation using silica gel column chromatography with the following solvent system: hexane: ethyl acetate (3:1), hexane: ethyl acetate (1:3), ethyl acetate, ethyl acetate: methanol (1:1), and methanol.

### 4.4. Cell Line

AGS cells (ATCC CRL-1739) and RAW cells (ATCC TIB-71) acquired at the Rio de Janeiro Cell Bank (BCRJ) were maintained in DMEM/nutrient mixture F-12 (DMEM/F12) and DMEM high glucose, respectively, supplemented with 10% of fetal bovine serum (Vitrocell, Campinas/SP, Brazil) and incubated at 37 °C with a 5% CO_2_ atmosphere.

### 4.5. Cytotoxicity Assay

AGS cells with 70–90% of confluence were trypsinized, counted in a Neubauer chamber, and 100 µL of the medium with approximately 1.2 × 10^5^ cells/mL was distributed on a 96-well plate and incubated at 37 °C and 5% CO_2_ atmosphere. After 24 h, the medium was removed and each well was treated with different concentrations of the *L. obtusa* samples—1, 10, and 100 µg/mL—in DMSO and medium. The concentration of the solvent in the wells was less than 0.05%. After 48 h of incubation, the MTT-tetrazolium assay was carried out [49]. A solution of 2% MTT-tetrazolium in PBS buffer was distributed on the 96-well plate previously treated with *L. obtusa* samples and, after 2 h, the plate was read in an ELISA Microplate Reader (iMark™ Bio-Rad, Hercules, CA, USA) at a wavelength of 540 nm. Cisplatin was used as cytotoxic standard and growth control was performed with the cells only (without treatment).

RAW cells, after they reached 70–90% of confluence, were gently harvested with a cell scraper, counted in a Neubauer chamber, and 100 µL of the medium with approximately 5 × 10^5^ cells/mL was distributed on a 96-well plate and incubated at 37 °C with a 5% CO_2_ atmosphere. After 24 h, the medium was removed and each well was treated with different concentrations of the Hex:AcOEt fraction—0.781, 1.562, 3.125, 6.25, 12.5, 25, 50, and 100 µg/mL—in DMSO and medium. The DMSO concentration in the wells was less than 0.05%. After 24 h of incubation, the MTT-tetrazolium assay was carried out as previously described [49].

### 4.6. Structural Determination

The mass spectrometry was carried out through electrospray ionization for the Hex:AcOEt fraction. The analysis was run in the positive and negative mode of ESI, with a range of 100–1000 *m*/*z*, using Fourier Transform Ion Cyclotron Resonance Mass Spectrometer (FT-ICR-MS, 9.4 T Solarix, Bruker Daltonics, Bremen, Germany). All mass spectra were calibrated with sodium trifluoroacetate (NaTFA) with a range of 200–1200 *m*/*z*. The average resolving power was m/Δm_50%_ = 52,170. The software Data Analysis 4.1 (Bruker Daltonics, Billerica, MA, USA) was used to identify the molecular formulas with the SmartFormula tool, in which C_15_ and C_20_ compounds were selected for the compound search, since these formulas suggest terpenes structures. The molecules suggested for each formula were assigned through the following databases: *Chemspider* (https://chemspider.com/, accessed on 22 September 2022), *PubChem* (https://pubchem.ncbi.nlm.nih.gov/, accessed on 26 September 2022), *KNApSAcK* (https://knapsackfamily.com/knapsack_core/top.php/, accessed on 27 September 2022), and *Seaweed Metabolite Database* (SWMD) (https://www.swmd.co.in/index.php/, accessed on 27 September 2022). The mass error was 50 ppm [50].

### 4.7. Statistics

The statistical analysis was performed with GraphPad Prism 5 software to verify the reproducibility and validate the results. A two-way analysis of variance (ANOVA) was used to verify statistically significant differences between the results of the samples and cytotoxic control treatment (cisplatin). Non-linear regression was used to calculate IC_50_.

### 4.8. Anticancer Biological Targets

The following biological target candidates for potential cytotoxic activity were pre-selected according to a bibliometric analysis: bcl-xl, HIF-1α, HIF-2α, cdc25, CDK4, and pRB. Afterwards, HIF-2α was selected as a biological target in this study considering physiological roles, crystallographic complex available, active sites, and known inhibitors.

### 4.9. Ligand Database

The chemical structures of the compounds assigned to the Hex:AcOEt were built in Marvin Sketch 21.10v software, minimized in Avogadro 1.2.0v, and saved as a .mol2 extension.

### 4.10. Crystallographic Complex

To determine the crystallographic complex of HIF-2α to be used in molecular docking, the 21 available complexes in Protein Data Bank (PDB) [51] (Appendix A) were inspected and analyzed with their ligands and their interactions in the active site of the protein (Appendix A). The 4GHI complex presented a better resolution in the visual inspection of the interactions between ligand and protein (Appendix A), and was therefore chosen for molecular docking. For further details, please access the Appendix A, and Figure 1).

### 4.11. Redocking

The validation of the docking model was obtained with the 4GHI ligand: 0X3. To determine the most suitable docking model, we analyzed the results of the four score functions available in the GOLD software—PLP, ASP, ChemScore, and GoldScore—with a radius ranging from 8 to 10 Å (Appendix A). The score function that presented more poses with root mean square deviation (RMSD) below 2 Å, the PLP 8 Å, was chosen for the docking model [52]. For further details, please access the Appendix A.

### 4.12. Molecular Docking

The GOLD software (Hermes 2020.3.0v CDCC) [53] was used to perform the molecular docking using the 4GHI PDB complex and the compounds found in Hex:AcOEt fraction. The calculations were executed following the redocking model: PLP score function with a radius of 8 Å. The protein and ligands were prepared according to the Hermes suite standards.

## 5. Conclusions

The results demonstrate that the Hex:AcOEt fraction presents significant and selective cytotoxic activity, which could be due to the presence of terpenes, an acetogenin, and a polyunsaturated fatty acid, as identified by mass spectrometry. Furthermore, from the molecular docking we can predict that substances found in the most cytotoxic fraction, such as chermesiterpenoid B, 11,12-dihydro-3-hydroxiretinol, and 7-hydroxy-10-dehydroxydeacetyldihydrobotrydial-1(10),5(9)-diene, showed cytotoxic activity for the first time in the literature, and inhibition potential at the HIF-2α active site as a possible antitumor target.

We know that this study is preliminary, but it shows that the Hex:AcOEt fraction from *L. obtusa* has potential to elucidate candidate natural products in the development of new antitumor drugs and should be evaluated by further studies; for example, by isolating and identifying the compounds suggested by the MS analysis. Furthermore, is necessary to evaluate the cytotoxicity of the isolated compounds and the effect on HIF-2α using PCR and Western blot.

## Figures and Tables

**Figure 1 marinedrugs-21-00318-f001:**
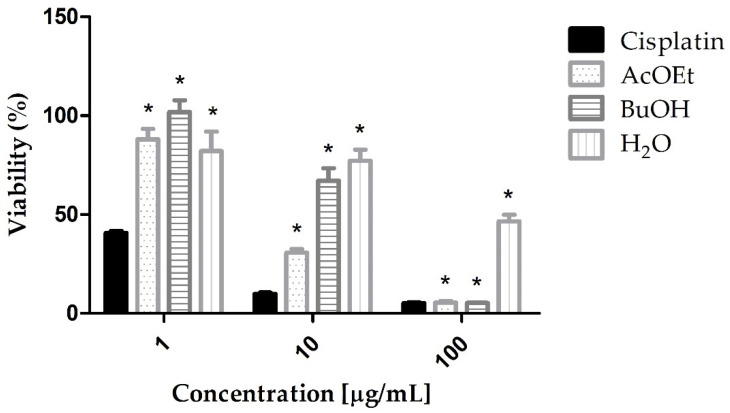
AGS cells viability after treatment with Cisplatin (cytotoxic standard), AcOEt, BuOH, and H_2_O fractions of *L. obtusa*. The data shown are the mean ± standard deviation of triplicate samples. The significance of the results was calculated by two-way ANOVA. Cell growth control corresponds to 100% viability. (*) *p* < 0.05.

**Figure 2 marinedrugs-21-00318-f002:**
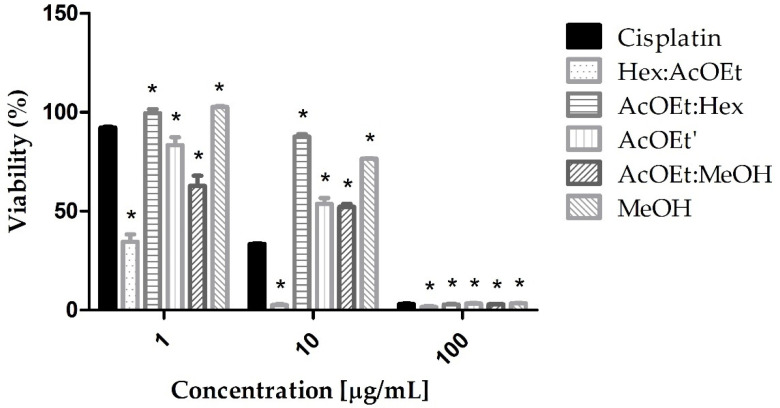
AGS cells viability after MTT assay with Cisplatin (cytotoxic standard), Hex:AcOEt, AcOEt:Hex, AcOEt’, AcOEt:MeOH, and MeOH fractions of *L. obtusa*. The data shown are the mean ± standard deviation of triplicate samples. The significance of the results was calculated by two-way ANOVA. Cell growth control corresponds to 100% viability. (*) *p* < 0.05.

**Figure 3 marinedrugs-21-00318-f003:**
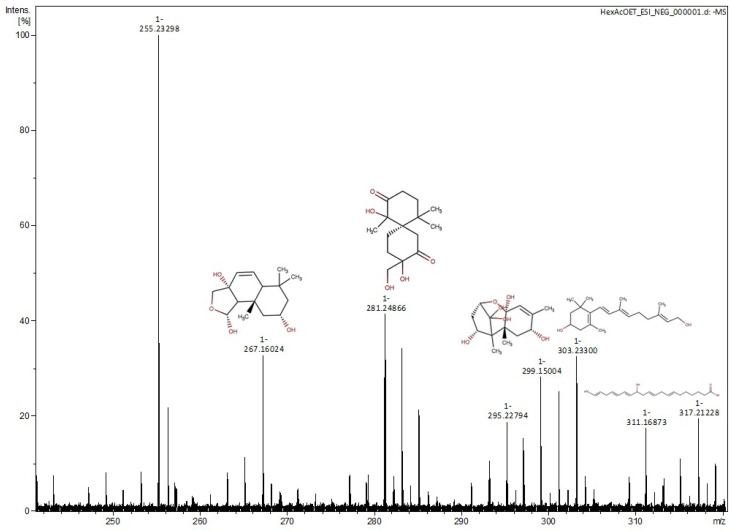
Mass spectra of Hex:AcOEt fraction. ESI-MS negative ion mode, 240–320 *m*/*z*.

**Figure 4 marinedrugs-21-00318-f004:**
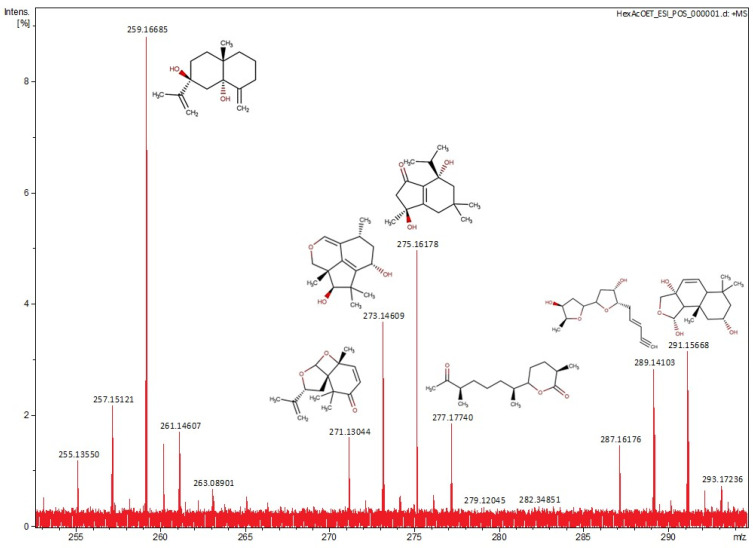
Mass spectra of Hex:AcOEt fraction. ESI-MS positive ion mode, 250–300 *m*/*z*.

**Figure 5 marinedrugs-21-00318-f005:**
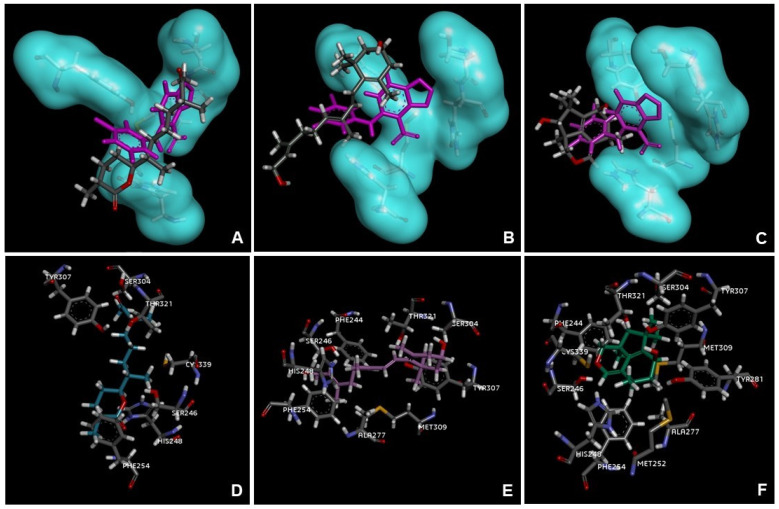
Interaction of the 0X3 reference ligand from HIF-2a crystallographic complex (PDB code 4GHI) and the compounds with the highest score in the docking calculations. The 0X3 ligand interacts hydrophobically with amino acids of the active site, changing their conformations (His248, Tyr281, Ser292, and especially Met252). (**A**–**C**) Chermesiterpenoid B, 11,12-dihydro-3-hydroxyretinol, and 7-hydroxy-10-dehydroxydeacetyldihydrobotrydial-1(10),5(9)-diene (carbon atoms in grey) overlaid with 0X3 (atoms in purple) interacting with the main amino acid residues of the protein active site, (**D**) Chermesiterpenoid B (carbon atoms in blue) interacting hydrophobically with amino acids residues Tyr307, Ser304, Thr321, Cys339, Ser246, His248, and Phe254 (carbon atoms in gray) of the protein active site, (**E**) 11,12-dihydro-3-hydroxyretinol (carbon atoms in purple) interacting hydrophobically with amino acids residues Tyr307, Ser304, Thr321, Phe244, Ser246, His248, Phe254, Ala277, and Met309 (carbon atoms in gray) of the protein active site, and (**F**) 7-hydroxy-10-dehydroxydeacetyldihydrobotrydial-1(10),5(9)-diene (carbon atoms in green) interacting hydrophobically with amino acids residues Phe244, Cys339, Ser246, His248, Phe254, Met252, Ala277, Tyr281, Met309, Tyr307, Ser304, and Thr321 (carbon atoms in gray) of the protein active site.

**Table 1 marinedrugs-21-00318-t001:** Suggested compounds in Hex:AcOEt fraction of *L. obtusa* from ESI-MS (−).

No.	*m*/*z*	Monoisotopic Mass	Formula	Compounds
1	267.1602	268.1677	C_15_H_24_O_4_	2α-hydroxy-7α,8α-epoxy-isodrimeninol [20]
2	283.1551	284.1623	C_15_H_24_O_5_	Merulinol F [21]
3	299.1500	300.1579	C_15_H_24_O_6_	Trichothecene 1 [22]
4	303.2330	304.2397	C_20_H_32_O_2_	11,12-dihydro-3-hydroxyretinol [23]
5	317.2123	318.4550	C_20_H_30_O_3_	9-hydroxy-2,5,7,11,14-eicosapentaenoic Acid [24]

**Table 2 marinedrugs-21-00318-t002:** Suggested compounds in Hex:AcOEt fraction of *L. obtusa* from ESI-MS (+).

No.	*m*/*z*	Monoisotopic Mass	Formula	Compounds
1	259.1668	236.1764	C_15_H_24_O_2_	Eudesma-4(15),11-dien-5,7-diol [25]
2	271.1304	248.1412	C_15_H_20_O_3_	Laureacetal C [26]
3	273.1461	250.1568	C_15_H_22_O_3_	7-hydroxy-10-dehydroxydeacetyldihydrobotrydial-1(10),5(9)-diene [27]
4	275.1618	252.1725	C_15_H_24_O_3_	5,9-dihydroxy-1(6)-brasilen-7-one [28]
5	277.1774	254.1882	C_15_H_26_O_3_	Chermesiterpenoid B [29]
6	289.1410	266.1518	C_15_H_22_O_4_	Laurefurenyne B [30]
7	291.1566	268.1677	C_15_H_24_O_4_	2α-hydroxy-7α,8α-epoxy-isodrimeninol [20]

**Table 3 marinedrugs-21-00318-t003:** Ranking of compounds scores of Hex:AcOEt fraction after molecular docking on the active site of HIF-2α.

Compound	Pose	Score
Chermesiterpenoid B	Dock 3	65.90
11,12-dihydro-3-hydroxyretinol	Dock 8	52.50
7-hydroxy-10-dehydroxydeacetyldihydrobotrydial-1(10),5(9)-diene	Dock 1	48.67
Laurefurenyne B	Dock 4	48.38
Eudesma-4(15),11-dien-5,7-diol	Dock 2	35.52

## Data Availability

Not applicable.

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
