# Peer review of "In Vitro and In Silico Evaluation of Red Algae Laurencia obtusa Anticancer Activity"

_marinedrugs, 2023, doi:10.3390/md21060318_

Round 1
Reviewer 1 Report (Previous Reviewer 1)
1. This paper does not have publication potential in its current stage.
2. There are several grammatical mistakes and wrongly written phrases present throughout the manuscript.
3. The purification and characterization of anticancer substances are not satisfactory, such as low intensities of MS spectra and the non-availability of proper methodology.
4. It seems the authors wrongly interpreted anticancer activity as the authors diluted the compound in different solvents and did not use those solvents as a control.
5. Overall there should be more clarification, more experiments, language improvement, and results are needed.
There are several grammatical mistakes and wrongly written phrases present throughout the manuscript.
Author Response
Responses to reviewer are attached.

Reviewer 2 Report (Previous Reviewer 2)
I marked (green) comment in the text.

Author Response
Thank you for your comment. We corrected and rephrased the sentence. We use word's track changes tool.
Reviewer 3 Report (New Reviewer)
In this submitted manuscript, authors reported the in vitro anticancer activity evaluation of the different fractions of the red algae Laurencia obtuse and the following in silico HIF-2α inhibitory evaluation of some compounds identified by the MS. It was important to disclose the potential cytotoxicity of some secondary metabolites of Laurencia obtuse, which have been not reported before.
However, it is recommended to be accepted after some revisions.
Comments:
1. As shown in Figure 1, AcOEt fraction exhibited weaker cytotoxicity than the H2O fraction at the concentration of 1 μg/mL, but greater than the latter at 10 μg/mL and 100 μg/mL, respectively. Why did a reversal occur?
2. What was the reason to divide the AcOEt fraction into five subfractions including hexane : ethyl acetate 3:1 (Hex:AcOEt), ethyl acetate : hexane 3:1 (AcOEt:Hex), ethyl acetate (AcOEt’), ethyl acetate : methanol 1:1 (AcOEt:MeOH) and methanol (MeOH)? Why not consider ethyl acetate 10:1 or 1:1? It is better to analyze and summarize the polarities and abundance of the reported key secondary metabolites of the alga Laurencia obtuse.
3. As shown in Figure 3, the most abundant peak at m/z 255.23298 was not assigned to a compound. Perhaps this compound exhibits significant cytotoxicity and needs to be further studied in silico evaluation.
4. How many active sites does the protein HIF-2α have? Why perform the molecular docking the ligand 0X3 conserved? It is better to give detail descriptions of the interactions of chermesiterpenoid B and the 11,12-dihydro-3-hydroxyretinol with the main amino acids in the active site of the protein HIF-2α overlaid with the ligand 0X3.
5. As mentioned in P6L151-153, ‘the sesquiterpenes 7-hydroxy-10-dehydroxydeacetyldihydrobotrydial-1(10),5(9)-diene, 5,9-dihydroxy-1(6)-brasilen-7-one, and the chermesiterpenoid B also have no evidence of citotoxycity in the literature’. It is as important to display the interactions of 7-hydroxy-10-dehydroxydeacetyldihydrobotrydial-1(10),5(9)-diene and 5,9-dihydroxy-1(6)-brasilen-7-one with the protein HIF-2α as that of chermesiterpenoid B.
6. In the experiment section, it was mentioned the following biological target candidates for cytotoxic activity were screened according to the literature: bcl-xl, HIF-1α, HIF-2α, cdc25, CDK4, and pRB. However, there were no data for the bioassay in the manuscript or Supplementary Materials.
Others:
1. Abstract: There was no unit for the selectivity index value 15.56, as estimated by the ratio of RAW/AGS IC50 values. And the compound scanlonenyne was not identified by the MS spectra in this manuscript.
2. P5L112&P7L189: The compound name ‘hydroxyretinol’ should be revised as ‘11,12-dihydro-3-hydroxyretinol’.
3. P6L152: ‘5,9dihydroxy-1(6)-brasilen-7-one’ → ‘5,9-dihydroxy-1(6)-brasilen-7-one’
4. As the discussion was vey long, it is better to add the Conclusions section in the manuscript.
5. The journal name was missing in Ref. [2].
There are some typo or grammar errors to be corrected, such as the following:
1. P2L70: ‘AcOEt has shown...’ → ‘The AcOEt fraction has shown...’
2. P3L96: ‘The mass spectrum of...’ → ‘The mass spectra of...’
3. P5L112: ‘The interaction of...’ → ‘The interactions of...’
4. P8L252: ‘0,05%’ → ‘0.05%’
5. P10L333: ‘H20’ → ‘H2O’
Author Response
Responses to the reviewer are attached.

Round 2
Reviewer 1 Report (Previous Reviewer 1)
1. Controls are missing in figure1 and figure2? Please provide the control details in figure legends and section text (2.1) as well. Also, update the figures with controls.
2. Section 2.1 should be improved and rewritten with all the relevant details.
3. Ethyle acetate, butanol, and water, all should be cytotoxic for the cells when used as control alone. Please justify, how the authors designed this experiment. Ideally, cells in only PBS or medium should be a standard control to show no cytotoxicity and compare with others. Please check this reference. https://www.nature.com/articles/srep46541
4. Please include a separate section showing the limitations of this manuscript.
This is fine.
Author Response
Please see the attachment.

Reviewer 3 Report (New Reviewer)
Although the authors answered all my concerns, but not all the corresponding revisions were made in the submitted manuscript.
Author Response
Corresponding revisions were made in the manuscript as suggested. Changes of this manuscript version are highlighted in yellow.
Round 3
Reviewer 1 Report (Previous Reviewer 1)
Limitation of the study.
#Authors need to include this section which is a must. In fact, there are several limitations with such natural compounds and experimental designs that should be described or at least discussed.
This is fine.
Author Response
We include the limitation of the study in the conclusion section. Changes are highlighted in yellow.
This manuscript is a resubmission of an earlier submission. The following is a list of the peer review reports and author responses from that submission.
Round 1
Reviewer 1 Report
1. Figure1. Use of water is not appropriate as the control? Did the authors perform the MTT assay in water? It seems the authors did not understand the logic of control here. Please justify.
2. Figure2. What are the negative and positive controls here? Please justify.
3. What is the purification status of the compounds? The authors need to present the purification data. There is no mean of this study if performed with unpurified compounds.
4. Authors need to show the cytotoxicity against normal cell lines also to confirm the safety.
5. The material methods section is weak, such as the docking section, with no details available.
6. How come the docking score comes in positive values? What is the reference for docking to say it is strong interaction? Please justify.
Reviewer 2 Report
I marked my comments in the text.
